# Transcriptome-Based Revelation of the Effects of Sleep Deprivation on Hepatic Metabolic Rhythms in Tibetan Sheep (*Ovis aries*)

**DOI:** 10.3390/ani14223165

**Published:** 2024-11-05

**Authors:** Ya-Le Chen, Ru Wang, Rui Pang, Zhi-Peng Sun, Xiao-Long He, Wen-Hui Tang, Jing-Yu Ou, Huan-Ming Yi, Xiao Cheng, Jia-Hong Chen, Yang Yu, Chun-Huan Ren, Qiang-Jun Wang, Zi-Jun Zhang

**Affiliations:** 1College of Animal Science and Technology, Anhui Agricultural University, Hefei 230036, China; chenyale1@163.com (Y.-L.C.); wangru3127@163.com (R.W.); ruina1991zx@163.com (R.P.); hexiaolong815@163.com (X.-L.H.); 23720297@stu.ahau.edu.cn (W.-H.T.); jy18027008632@163.com (J.-Y.O.); yihuanming05@163.com (H.-M.Y.); chengxiao@ahau.edu.cn (X.C.); renchunhuan@ahau.edu.cn (C.-H.R.); 2Chongqing Key Laboratory of Herbivore Science, College of Animal Science and Technology, Southwest University, Chongqing 400715, China; fsunzhipeng@163.com; 3Center of Agriculture Technology Cooperation and Promotion of Dingyuan County, Chuzhou 233200, China; chenjiahong@ahau.edu.cn; 4Qinghai Provincial Key Laboratory of Adaptive Management on Alpine Grassland, Qinghai Academy of Animal Science and Veterinary Medicine, Qinghai University, Xining 810016, China; greencatqi@126.com

**Keywords:** sleep deprivation, circadian rhythm, liver, lipid metabolism, Tibetan sheep

## Abstract

Sleep is a vital physiological process essential for maintaining mental and physical health. Insufficient sleep affects various physiological processes, including liver function and metabolism. In this study, we used a Tibetan sheep model for the first time to investigate the effects of sleep deprivation (SD) for 7 days (6 h/day) on liver function and circadian rhythms. We found that SD significantly disrupted the expression patterns of circadian rhythm genes in the liver. Gene enrichment analysis further indicated alterations in pathways related to lipid metabolism in the liver, suggesting that SD resets the expression of metabolic genes, causing liver damage by disrupting lipid metabolism.

## 1. Introduction

Sleep is an important physiological process associated with essential physical and mental health functions [1]. Approximately one-third of our lives is spent sleeping [2]. However, the average human sleep duration is decreasing owing to modern lifestyle changes, such as shift work, nighttime social activities, and heavy academic pressure on teenagers [3,4,5]. Epidemiological surveys have found that children and adolescents worldwide do not obtain enough sleep, and the percentage of working adults sleeping less than 7 h per day is increasing globally [6]. Meanwhile, the “China Sleep Research Report (2022)” found that the average sleep duration of Chinese decreased from 8.5 h in 2012 to 7.06 h in 2021 [7]. Insufficient sleep increases the risk of obesity, diabetes, cardiovascular disease, and other metabolic diseases in humans [8,9]. Even mild SD can significantly affect cognitive function, mental and physical health, work performance, and safety [10]. Similarly, adequate sleep is essential for livestock production, where animals can maintain high behavioral performance and low-stress responses in a good resting environment. Insufficient sleep increases levels of stress hormones and disease susceptibility, reduces immune function, and reduces production performance in animals [11,12]. Previous studies have revealed that cows should rest for 12–13 h daily, and deprivation of resting time can lead to stress responses such as elevated adrenocorticotropic hormone and reduced plasma concentration of growth hormone, resulting in decreased milk production [13,14,15]. Transportation and high stocking densities can affect the rest and sleep of animals, thereby affecting productivity and posing significant challenges to animal health and welfare.

The liver, serving as the metabolic hub in the human body, plays a crucial role in various aspects of metabolism, including carbohydrate, fat, and protein metabolism [16]. Gluconeogenesis from propionate in the liver is the primary pathway for glucose synthesis in the body. Dysregulation of nutrient metabolism in the liver can contribute to chronic metabolic disorders [17,18]. Previous studies have revealed that some cows are prone to fatty liver disease and other lesions during early lactation [19]. The biological clock can control these physiological activities in the liver, and disruptions in biological rhythms can cause disturbances in liver metabolic functions, accelerating the development of liver diseases, such as fatty liver, cirrhosis, hepatitis, and liver cancer [20,21,22]. SD can disrupt the natural sleep–wake cycle, causing circadian rhythm disturbances affecting various physiological processes in the body, including liver function and metabolism. Previous studies have found that SD in rodents reduces endogenous antioxidant defenses in the liver, and long-term SD can cause oxidative damage in visceral organs [23,24]. The sleep pattern and feeding behavior of mice undergo changes within a span of two days due to sleep deprivation, while prolonged sleep deprivation significantly impacts the rhythmic expression of genes associated with carbohydrate, lipid, and protein metabolism pathways in the liver of mice [25]. He et al. found that sleep restriction affected lipid metabolism in the liver and gut microbiota and promoted cholesterol gallstone formation in mice. Moreover, the addition of the oligopeptide SEP-3 can achieve liver injury repair by inhibiting hepatocyte apoptosis, activating the hepatic Wnt/β-Catenin pathway, and promoting hepatocyte proliferation and migration [26]. Adding melatonin to diets partially restored hepatic lipid metabolism homeostasis in mice fed a high-fat diet at night [27]. Furthermore, melatonin supplementation can increase the 5′ AMP-activated protein kinase (AMPK)α/peroxisome proliferator-activated receptor (PPAR)α signaling pathway activity in the liver, leading to lipid breakdown, reducing fat accumulation, and preventing weight gain due to sleep deprivation in mice [28]. Previous studies have investigated circadian rhythms in the liver using high-throughput transcriptomics, proteomics, and lipidomics [29,30,31,32]. However, studies on the effects of SD on liver circadian rhythms are relatively scarce and have been primarily conducted on nocturnal animals, such as rodents, with no reports on large diurnal animals.

The use of sheep as models to study human diseases has increased significantly over the past two decades [33]. Many studies have explored the relationship between brain function and metabolic activity using the sheep model because sheep are a daytime animal, and their brain, life span, and rhythm habits are closer to humans than rodents [34]. Because the physiology of the sheep conceptus is similar to that of the human fetus, the sheep model has also been widely used in studies of fetal development and physiology. Some studies have used sheep models to explore the effects of maternal malnutrition on fetal liver, and the results show that malnutrition can cause serious lipid metabolism disorders and oxidative stress, thus affecting the normal development of fetal liver [35]. The present study developed a sleep deprivation model for Tibetan sheep, wherein the animals were subjected to 7 days of sleep deprivation (6 h per day). Liver tissues were then sequenced at 4 h intervals over a complete light cycle (24 h) using the high-throughput sequencing platform Illumina. This approach aimed to investigate the impact of sleep deprivation on liver metabolic rhythms in sheep. Our findings shed light on the intricate interactions between sleep patterns and liver metabolic processes, providing valuable insights into mitigating rest and sleep deprivation resulting from improper transportation and management practices in livestock production.

## 2. Materials and Methods

### 2.1. Ethics Statement

All animal management and experimental procedures were performed according to the Guidelines for Experimental Animals established by the Ministry of Science and Technology of China. All experimental protocols were approved by the Institutional Animal Care and Use Committee (IACUC) of Anhui Agricultural University (approval number AHAUXMSQ2024175).

### 2.2. Experimental Design and Sample Collection

We selected 60 male Tibetan (*Ovis aries*) lambs (6-month-old sheep, initial body weight 23.1 ± 0.5 kg) and raised them in the sheep house of the Hangzhou Animal Husbandry Co., Ltd., Hangzhou, China. All Tibetan sheep were housed in separate cages (five per cage) and had ad libitum access to water. The sheep were fed twice daily, at 7 am and 5 pm. The residual feed was weighed, and the cages were cleaned daily. The average daily feed intake was recorded. The feed was free of antibiotics and met the standard nutritional needs of growing Tibetan sheep (Appendix A). The sheep were acclimatized to the environment for 1 week before the trial began. All sheep were kept under 12 h light/dark conditions (ZT: zeitgeber time; ZT0 represents sunrise at 6 am). Sheep were randomly divided into a normal-sleep control group (COS, n = 30) and an SD model group (SDS, n = 30). The COS group had no sleep interventions, whereas the SDS group was sleep-deprived from ZT18 to ZT0 (Figure 1A). After 1 week of SD, liver and serum samples were collected every 4 h over a 24 h cycle (n = 4). Both groups were raised under identical temperature and humidity conditions (Appendix A). There were no significant differences in the daily feed intake or body weight changes between the two groups before and after the trial (Figure 1B).

### 2.3. Sleep Deprivation

COS and SDS Tibetan sheep were housed in two separate sheep houses to ensure that the behaviors of the two groups of sheep did not affect each other. The sheep houses had controlled environmental conditions (artificial lights and lights on and off at 7 am and 7 pm, respectively). Each group was housed in six pens (five sheep per pen), and infrared cameras were installed above each pen to monitor and record sheep behavior. The SD model was designed considering that used in previous studies [36]. During the SD period (ZT18–ZT0), sheep were observed every 5 min by walking around the sheep house. When the sheep exhibited signs of drowsiness, the sheep would be woken or disturbed by approaching the pen. If not woken up, the experimenter would enter the pen and gently wake the sheep by touching them. For the next 24 h, the sheep were not disturbed except for regular feeding activities.

### 2.4. Histological Analyses

Tibetan sheep liver tissues were removed and fixed in a formaldehyde solution (4%) for 48 h. Fragments were dehydrated in a graded ethanol series, embedded in paraffin, and sectioned using an automatic microtome at 4–5 mm thick. A terminal deoxynucleotidyl transferase dUTP nick-end labeling (TUNEL) assay kit (Sigma-Aldrich, Munich, Germany) was used to evaluate the apoptosis rate in Tibetan sheep hepatocytes. The TUNEL assay was performed using the TUNEL kit according to the manufacturer’s instructions. The lipid content of the Tibetan sheep liver was detected using Oil Red O staining (Sigma-Aldrich, Munich, Germany). The liver tissues were embedded in Tissue-Tek Optimal Cutting Temperature Compound (Sakura Europe, Leiden, Netherlands) and flash-frozen in cold isopentane. Next, 5 µm thick tissue sections were stained with Oil Red O staining for lipid content analysis. The mean droplet size was quantified using ImageJ software (version 1.50a; National Institutes of Health, Bethesda, MD, USA).

### 2.5. Measurement of Serum Biochemistry

The total cholesterol (TC) and triglyceride (TG) levels in Tibetan sheep serum were determined using a total cholesterol assay kit (Nanjing Jiancheng Bioengineering Institute, Nanjing, China) and a TG assay kit (Nanjing Jiancheng Bioengineering Institute, Nanjing, China), respectively. The serum levels of high-density lipoprotein (HDL) and low-density lipoprotein (LDL) were determined using a direct method. The alanine aminotransferase (ALT) and aspartate aminotransferase (AST) contents in serum were measured using an alanine aminotransferase assay kit (Jiangsu Meimian Industrial Co., Ltd., Nanjing, China) and an aspartate aminotransferase assay kit (Jiangsu Meimian Industrial Co., Ltd.), respectively.

### 2.6. Liver RNA Preparation and Assay

The liver tissue samples were collected for RNA extraction, sequencing, and library construction. Total RNA was extracted using TRIzol reagent (Invitrogen, Carlsbad, CA, USA). RNA purity was assessed using agarose gel electrophoresis and a NanoDrop 2000 spectrophotometer (Thermo Fisher Scientific, Waltham, MA, USA) (OD260/280 and OD260/230 ratios), and RNA integrity was precisely detected using an Agilent 2100 Bioanalyzer (Agilent, Santa Clara, CA, USA). Paired-end sequencing was performed using an Illumina NovaSeq X Plus (Illumina, San Diego, CA, USA).

### 2.7. Bioinformatic Analysis

The raw sequence data have been deposited in the NCBI Sequence Read Archive (SRA) under the accession number [PRJNA1177242] [37]. The reads were aligned to the ribosomal RNA (rRNA) database using Bowtie2 (version 2.2.8 [GitHub, San Francisco, CA, USA]) to remove rRNA-mapped reads and obtain clean reads [38]. An index of the reference genome was constructed, and clean paired-end reads were mapped to the reference genome using HISAT2 2.1.0 (GitHub) with default settings [39]. The mapped sequences for each sample were assembled following the reference method using StringTie v1.3.1 (National Institute of Health, Bethesda, MD, USA) [40,41]. Expression abundance and variation were determined using RNA-Seq using Expectation–Maximization (GitHub) by calculating the fragments per kilobase of transcript per million mapped reads for each transcription region [42].

### 2.8. Real-Time Qualitative Polymerase Chain Reaction Analysis

Reverse-transcribed total RNA was analyzed for mRNA expression using the SoFast EvaGreen Supermix on a CFX96, a real-time quantitative polymerase chain reaction machine (Bio-Rad, Hercules, CA, USA). The relative expression levels of mRNA were standardized to glyceraldehyde-3-phosphate dehydrogenase, and data analysis was performed using the 2^−ΔΔCT^ method (primer pairs are listed in Appendix A).

### 2.9. Statistical Analysis

Gene differential expression analysis was performed using EdgeR software (version 3.2) (Bioconductor, Boston, MA, USA), and genes with a fold change >1.5 and FDR < 0.05 were selected as differentially expressed genes (DEGs). The identification of oscillating genes was based on Jonckheere−Terpstra−Kendall (JTK) analysis, and genes exhibiting significant circadian rhythmic changes had an ADJ.P < 0.05 [43]. The expression pattern clustering analysis of oscillating genes was conducted using the Mfuzz fuzzy c-means algorithm in R (RStudio, Boston, MA, USA). Gene ontology and Kyoto Encyclopedia of Genes and Genomes (KEGG) enrichment analyses of significantly rhythmic and differentially expressed genes were performed using DAVID bioinformatics software (https://david.ncifcrf.gov/ accessed on 1 September 2024). Gene Set Enrichment Analysis (GSEA) bioinformatics software (https://www.broadinstitute.org/gsea/ accessed on 1 September 2024) was used for the enrichment analysis of differentially expressed genes.

## 3. Results

### 3.1. Sleep Deprivation Alters the Circadian Rhythm of Gene Expression in the Liver

To evaluate the impact of SD on the circadian rhythm of gene expression in the sheep liver, high-throughput transcriptome sequencing was performed on liver samples from two groups of sheep, identifying 21,529 expressed genes. A total of 40,596,236 raw reads were obtained. After screening, 39,015,032 clean reads were obtained (Appendix A). JTK analysis revealed that 3605 (16.85%) and 2847 (13.30%) genes in the COS and SDS groups, respectively, exhibited circadian rhythms (Figure 2A). Both groups had 709 genes exhibiting circadian rhythms (Figure 2B). Radar plots and clustered heat maps exhibited that the oscillating genes in both groups were distributed throughout the day, with peak values in the COS group appearing primarily at 8:00 (ZT2), 22:00 (ZT16), and 0:00 (ZT18), whereas the SDS group exhibited no significant peaks (Figure 2C,D). KEGG enrichment analysis of these genes (ADJ.P < 0.05) revealed that circadian rhythm genes in the COS group were enriched in pathways such as steroid biosynthesis, PPAR, AMPK, apelin (APJ) signaling pathways, and fatty acid metabolism and degradation, while circadian rhythm genes in the SDS group were mainly enriched in steroid biosynthesis, complement and coagulation cascades, and B cell receptor signaling pathways (Figure 2E). We further analyzed the genes with significant circadian rhythms (ADJ.P < 0.01) in both groups and found that 92.04% (ADJ.P < 0.01) of the genes in the sheep liver lost their circadian rhythms after SD treatment (Appendix A), as visually illustrated in the clustered heat map in Appendix A, indicating that SD affects gene expression in the liver by altering the circadian rhythm of expressed genes.

### 3.2. Sleep Deprivation Alters the Expression Patterns of Circadian Rhythm Genes in the Liver

To further elucidate the role of oscillating genes in regulating physiological processes in sheep, the fuzzy c-means algorithm was applied to cluster the expression levels of rhythmic genes. The results revealed that three distinct temporal expression clusters were identified in the COS and SDS groups, indicating different dynamic characteristics of rhythmic gene expression (Figure 3A). KEGG analysis of different clusters revealed that cluster 1 in the COS group was mainly enriched in pathways related to steroid synthesis, glycolysis/gluconeogenesis, and glutathione metabolism; cluster 2 was mainly enriched in pathways related to ribosome biogenesis in eukaryotes, mRNA surveillance, and cytoplasmic transport; and cluster 3 was mainly enriched in pathways related to the NF-κB signaling pathway, platelet activation, NOD-like receptor signaling pathway, and glucagon signaling pathway. Contrastingly, cluster 4 in the SDS group was mainly enriched in immune-related pathways such as complement and coagulation cascades, platelet activation, and endocytosis. Cluster 5 was mainly enriched in pathways such as proteasomes and ATP-dependent chromatin remodeling. Cluster 6 was mainly enriched in pathways such as steroid synthesis, pentose phosphate pathway, and NF-κB signaling pathway (Figure 3B), suggesting that SD disrupts the diurnal rhythms of gene expression in sheep liver.

### 3.3. Sleep Deprivation Alters Metabolic Pathways in Tibetan Sheep Liver

A comparative analysis of expressed genes between the COS and SD treatment groups was conducted to further elucidate the role of expressed genes in regulating physiological processes in the sheep liver, resulting in the identification of 106 differentially expressed genes (DEGs). In comparison to the COS group, 52 DEGs were found to be upregulated, while 54 DEGs were downregulated in the SDS group (Appendix A). KEGG analysis of the upregulated and downregulated genes revealed that the upregulated genes were mainly enriched in pathways such as circadian rhythm, estrogen signaling pathway, MAPK signaling pathway, longevity regulating pathway-multiple species, antigen processing and presentation, and lipid and atherosclerosis, whereas the downregulated genes were mainly enriched in ovarian steroidogenesis, steroid hormone biosynthesis, and cytosolic DNA-sensing pathway (Appendix A).

JTK analysis of these DEGs revealed that 21.70% and 20.75% of the DEGs in the COS and SDS groups, respectively, exhibited circadian rhythms (Figure 4A), with four DEGs exhibiting circadian rhythms in both groups (Figure 4B). GSEA of lipid metabolism and apoptosis genes revealed that gene sets related to lipid metabolism were generally expressed at high levels in the COS group than that in the SDS group. Simultaneously, the expression of the genes involved in apoptosis was higher in the SDS group than that in the COS group (Figure 4C), suggesting that SDS may alter the circadian rhythm of gene expression in sheep livers by affecting lipid metabolism.

### 3.4. Impact of SD on Serum Biochemical Indicators, Lipid Metabolism, and Apoptosis in Tibetan Sheep Liver

To further validate the effect of SD on lipid metabolism in the sheep liver, blood was collected at ZT2 to measure the ALT, AST, TG, TC, high-density lipoprotein, and low-density lipoprotein levels in sheep plasma. Liver tissue samples were collected at ZT2 for Hematoxylin and eosin, Oil Red O, and TUNEL staining. The results exhibited that SD significantly increased the plasma AST and ALT levels compared with those of the COS group (*p* < 0.05). Furthermore, the TUNEL assay revealed that the number of SD-positive cells was significantly higher than that in the COS group (Figure 5A). The results of serum biochemical indicators related to lipid metabolism exhibited that, compared with those of the COS group, SD significantly increased the TC, TG, and HDL activities in sheep blood (*p* < 0.05), while the level of LDL was significantly reduced (*p* < 0.05) (Figure 5A), indicating that liver damage induced by SD may be related to lipid metabolism disorders. To further study the effect of SD on liver fat accumulation in sheep, Oil Red O staining of liver tissue was performed, which exhibited that the lipid droplet content in the sheep liver in the SD group was significantly higher than that in the COS group (Figure 5C,D), suggesting that SD affects lipid metabolism in sheep, leading to fat accumulation in the liver and increased liver cell apoptosis, resulting in liver damage.

At the same time, we randomly selected five genes (TNFSF10, ENDOG, ACSL3, LIPG, and LPCAT3) from the differentially expressed genes and conducted RT-qPCR validation to investigate the impact of SD on hepatic gene expression in Tibetan sheep. Our findings revealed that SD intervention resulted in a downregulation of TNFSF10, ENDOG, and ACSL3 expression in the liver, while it upregulated LIPG and LPCAT3 expression in this organ (Figure 5E). These results are consistent with the observed trends from RNA-seq analysis.

## 4. Discussion

Sleep is one of the most important components of biological life activities, and sleep quality and duration are closely related to physiological and psychological health. To investigate the effects of sleep deprivation on hepatic metabolic rhythms, we conducted a 7-day (6 h/day) sleep deprivation regimen in Tibetan sheep and analyzed their hepatic transcriptome using RNA-seq. Our analysis revealed that sleep deprivation effectively resets the circadian rhythmicity of expressed genes in sheep liver, resulting in altered expression of rhythmic genes, accompanied by changes in metabolic pathways, including lipid metabolism, iron death, and apoptotic pathways. Notably, sleep deprivation disrupted normal sleep–wake rhythms in sheep liver, resulting in liver lipid metabolism disorder and increased liver cell apoptosis.

The liver, a peripheral organ with a highly responsive circadian rhythm, exhibits rhythmicity in several key functions [44]. Studies have found that approximately 15% of the liver transcriptome has circadian rhythms. When detecting the expression levels of organ genes in a concentrated background of 19,788 known protein-coding mouse genes using the JTK CYCLE (*q*-value < 0.05) algorithm, 3186 genes in the mouse liver exhibited circadian rhythmic expression [45], which is consistent with the findings from this study, where we detected 21,529 expressed genes through transcriptome sequencing of the sheep liver and 3605 (16.85%) genes exhibited circadian rhythms using the JTK CYCLE algorithm. Previous studies have found that the expression peaks of genes controlled by circadian rhythms in mouse organs are distributed throughout the day/night cycle, which is consistent with our findings. However, Satchidananda et al. found that the largest cluster of circadian rhythm-regulated transcripts in the mouse liver appeared at CT14 (20:00) and CT6 (12:00), whereas our study found that the peaks of circadian rhythm-regulated transcripts in the sheep liver mainly appeared at ZT2, ZT16, and ZT18, suggesting that liver-mediated physiological processes have unique temporal structures for each individual, with differences in the temporal organization of the liver between nocturnal (mice) and diurnal (sheep) animals [46]. Our study investigated the KEGG enrichment analysis of genes related to the circadian rhythm in the sheep liver circadian rhythm map and found that circadian rhythm genes in the sheep liver were mainly enriched in pathways such as steroid biosynthesis, PPAR, AMPK, and APJ signaling pathways, as well as fatty acid metabolism and degradation, consistent with the findings of Artem et al. [47]. After SD, the enrichment pathways of the circadian rhythm genes in sheep liver were significantly different from those in the control group. The circadian rhythm genes in the SDS group were enriched in steroid biosynthesis, complement and coagulation cascades, and B cell receptor signaling pathways. Liu et al. found that long-term SD can induce circadian rhythm reprogramming of the liver metabolic transcriptome, significantly changing the rhythmic expression of genes related to carbohydrate, lipid, and protein metabolism pathways [25], indicating that SD alters the transcriptional profiles of circadian rhythm genes in the liver.

SD disrupts circadian rhythms and affects gene expression patterns in the liver. The Mfuzz analysis tool identified and categorized different gene expression patterns in the time-series data [48]. Huang used Mfuzz to cluster the expression data of different oscillating genes in a study on the circadian transcriptome of the retina under SD and found that SD disrupted the intrinsic rhythms of genes and affected gene expression patterns in the retina [49]. A previous study that sequenced the liver transcriptome of sleep-deprived mice found that most genes involved in pathways such as fatty acid metabolism did not have circadian rhythms under normal conditions; however, after prolonged SD, similar rhythmic expression appeared in two troughs at ZT5 and ZT6, indicating that SD resets the rhythmicity of multiple metabolic pathways while affecting liver gene expression patterns [25]. In this study, we found that the expression patterns of circadian rhythm-related genes in the sheep liver changed significantly after SD.

Lack of sleep causes circadian rhythm and liver metabolic disorders, particularly lipid metabolism imbalances [50]. In a study, it was found that compounds in the model associated with insufficient human sleep are related to factors such as lipid homeostasis [51]; simultaneously, when studying sleep onset time, multiple associations were discovered between sleep timing and lipid metabolism products [50]. Many enzymes regulating lipid metabolism exhibit rhythmic oscillations. For example, previous studies have found that two key enzymes involved in sphingolipid metabolism, UGCG and SPTLC2, exhibit rhythmic circadian expression in human skeletal muscle tissue [52]. Furthermore, a previous lipidomic study of the mouse liver found that approximately 17–50% of the lipid content exhibited circadian rhythmicity [53], indirectly suggesting a correlation between changes in the expression of lipid-regulating enzymes and rhythmic changes, consistent with the findings of our study, where many rhythmically changing genes regulated processes such as lipid metabolism. Apelin is an adipokine that acts on the G protein-coupled receptor APJ. Studies have found that Apelin–APJ signaling in hepatocytes protects receptor-mediated steatosis in human and mouse hepatocytes via AMPK and PPARα [54], consistent with findings from this study that revealed that the enrichment of circadian rhythm genes in pathways such as the PPAR, AMPK, and APJ signaling pathways is lost in the sheep liver after SD. Additionally, SD elevates glyceride and glycerophospholipid levels in the human blood, which may be related to abnormal liver lipid metabolism and fatty lesions, as consistent with the findings of other studies [55]. For example, a previous study on the nuclear receptor Rev-erbα, which regulates liver lipid metabolism and circadian rhythms, found that Rev-erbα maintains the activity levels of transcription factors involved in cholesterol metabolism [56]. Meanwhile, Rev-erbα can rhythmically recruit histone deacetylases (HDACs), affecting the expression of key enzymes such as acyl-CoA Synthetase long-chain family member 3 (ACSL3) and Lysophosphatidylcholine Acyltransferase 3 (LPCAT3) in the TG metabolism pathway, thereby participating in glycerol and TG metabolism [57]. Furthermore, Rev-erbα knockout mice exhibit liver lipid droplet accumulation, hypertrophy, and fatty lesions [58], indicating that rhythm disorders caused by SD may affect liver fat accumulation, degradation, and related gene expression changes through pathways, including the mechanisms mentioned above. In this study, transcriptome analysis revealed that key genes regulating ferroptosis, such as Glycerol-3-Phosphate Acyltransferase 3/lipase G and LPCAT3, were upregulated, whereas ACSL3 was downregulated, and differentially expressed genes were enriched in the ferroptosis pathway, consistent with the findings of previous studies; however, the specific molecular mechanisms need further investigation.

SD affects hepatocytes and leads to liver cell damage, exacerbating apoptosis and liver injury, with hepatocyte apoptosis being a pathogenic factor in certain liver diseases [59]. Researchers, including Jie Yang, have found that shortened sleep duration increases the risk of non-alcoholic fatty liver disease (NAFLD)/metabolic associated fatty liver disease (MAFLD) [60]. Furthermore, sleep disruption, including short nighttime sleep and longer daytime naps, further exacerbates the risk of NAFLD in middle-aged and elderly individuals [61]. Lu Liu et al. found that SD increases the number of TUNEL-positive hepatocytes in the liver tissue of mice exposed to LPS/D-Gal, which is consistent with our data [62]. In our study, the SD group showed more TUNEL-positive hepatocytes, and transcriptome results revealed that differentially expressed genes after SD were enriched in pathways related to apoptosis. In the apoptosis pathway, the expression of TNFSF10 was significantly higher in the SD group than in the control group. TNFSF10 is a member of the TNF family, and TNF-α has a potent pro-apoptotic effect. In the LPS/D-Gal model, TNF-α is the main mediator of liver injury [63,64]. Therefore, in the Tibetan sheep model, SD may exacerbate liver injury by promoting hepatocyte apoptosis. Some studies have found that sleep deprivation can cause varying degrees of damage to the body through ferroptosis [65]. For instance, sleep deprivation in pregnant rats led to hippocampal neuroinflammation damage in the offspring, mediated by ferroptosis [66]. Abnormal lipid metabolism may enhance lipid peroxidation and induce ferroptosis by altering the lipid composition of biological membranes [67]. Studies have shown that low levels of PUFA-PL (polyunsaturated fatty acid–phospholipid) can enhance resistance to ferroptosis in activated neutrophils [68]. ACSL4 and LPCAT3 are key mediators in PUFA-PL synthesis [69], and ACSL4 and ACSL3, two isoforms of ACSLs, are critical for ferroptosis [70]. These findings suggest that sleep deprivation-induced ferroptosis may cause liver damage by affecting the aforementioned mechanisms and related gene expression changes. In this study, transcriptome analysis revealed that LPCAT3, a key gene regulating ferroptosis, was upregulated, while ACSL3 was downregulated, and differential genes were enriched in the ferroptosis pathway. These results align with previous studies [71].

## 5. Conclusions

To our knowledge, this is the first study using a Tibetan sheep model to investigate the effects of SD on the liver. Our study revealed that SD significantly disrupted the circadian rhythms of the liver in Tibetan sheep, leading to altered gene expression and changes in liver metabolism. Transcriptome analysis of sheep liver exhibited that SD can induce rhythmic changes in many genes, particularly those related to lipid metabolism, apoptosis, and ferroptosis pathways. These changes were validated at the levels of blood biomarkers, liver tissue staining, and gene expression. These findings provide new insights into the effects of SD on the liver circadian rhythms and their potential mechanisms, highlighting the importance of sleep in maintaining metabolic homeostasis. Future studies should explore the long-term consequences of SD on liver function and overall health and the potential correlations among SD, circadian rhythm disruption, and metabolic disorders. Our findings elucidate the complex interactions between sleep patterns and liver metabolic processes, providing insights into combating rest and SD caused by improper transportation and management of livestock production. The results of this study offer new perspectives for improving sheep transportation and addressing the reduced productivity caused by poor daily management practices.

## Figures and Tables

**Figure 1 animals-14-03165-f001:**
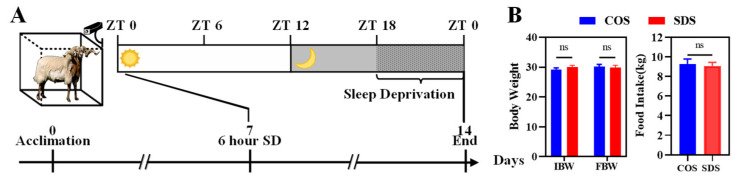
Experimental design and daily weight gain of Tibetan sheep (*Ovis aries*). (**A**) Schematic diagram of the experimental design. (**B**) Food intake and body weight changes in different treatment groups (*t* test, ±SEM, n = 30 per group). SD, sleep deprivation; ZT, zeitgeber time. ns means no significant difference.

**Figure 2 animals-14-03165-f002:**
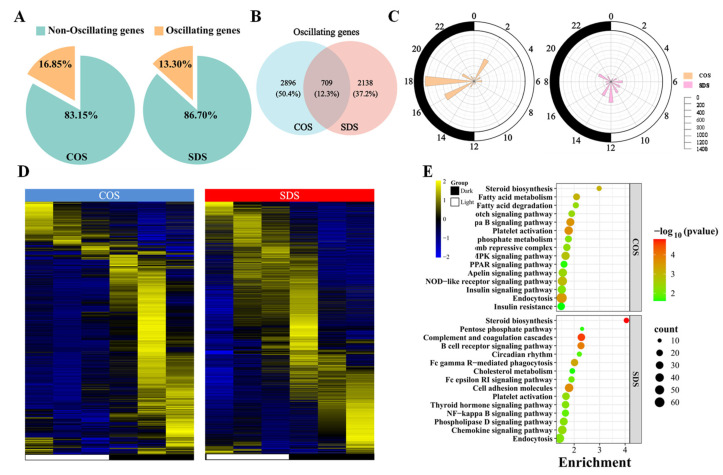
Sleep deprivation caused a change in the circadian rhythm of the genes expressed in the liver of Tibetan sheep. (**A**) Pie chart diagram showing the percentage of oscillating and non-oscillating genes in the COS and SDS groups (ADJ.P < 0.05). (**B**) Venn diagram displaying the number (top) and ratio (bottom) of oscillating genes in liver samples isolated from Tibetan sheep in the COS and SDS groups (ADJ.P < 0.05). (**C**) The polar plot represents time when the genes’ peak level of abundance appeared; yellow (COS group) or pink (SDS group) shading represents the number of rhythmic genes with an estimated peak value for each time as determined by JTK analysis. The radius of black concentric circles indicates the number of rhythmic genes, and the minimum radius of the black concentric circle represents 200 genes. The black arc on the left side of the polar plot indicates the day/night cycle. (**D**) Heat map showing oscillating genes in the COS (left) or SD (right) groups (ADJ.P < 0.05). (**E**) KEGG-enriched pathways for the expression of COS group oscillating genes and SDS group oscillating genes (*p* < 0.05).

**Figure 3 animals-14-03165-f003:**
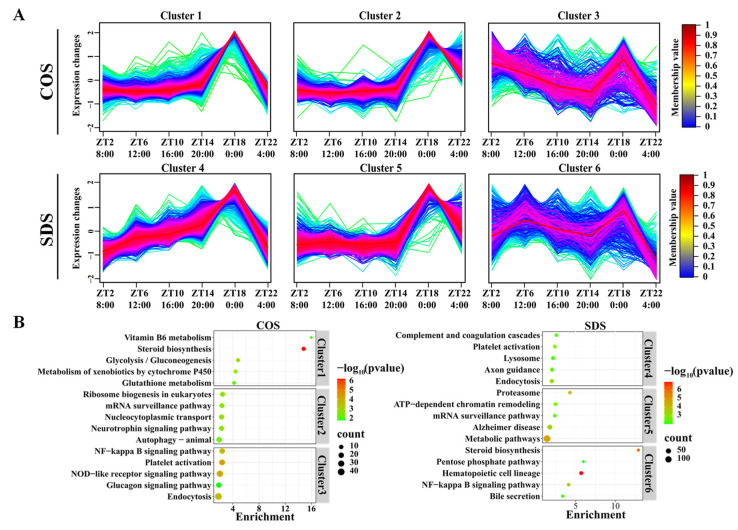
Sleep deprivation alters the diurnal pattern of rumen metabolites in Tibetan sheep. (**A**) Identification of distinct temporal patterns of expression genes in the COS and SDS groups by fuzzy c-means clustering. The *y*-axis represents normalized data based on all expression genes within each cluster. (**B**) KEGG analysis (*p* < 0.05) of the metabolites within each cluster in the COS and SDS groups.

**Figure 4 animals-14-03165-f004:**
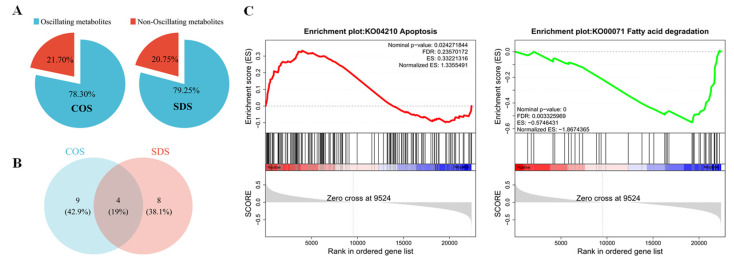
Sleep deprivation alters the circadian rhythm of the liver in Tibetan sheep. (**A**) Pie chart diagram showing the percentage of oscillating and non-oscillating DEGs in the COS and SDS groups (ADJ.P < 0.05). (**B**) Vene diagram displaying the number (top) and ratio (bottom) of oscillating DEGs in liver samples isolated from Tibetan sheep in the COS and SDS groups (ADJ.P < 0.05). (**C**) The lipid metabolism and apoptosis-related pathways derived from the GSEA analysis of the liver gene profiling (|NES| > 1, NOM *p*-val < 0.05, FDR q-val < 0.25). The red line indicates high expression in the SDS group, and the green line indicates high expression in the COS group.

**Figure 5 animals-14-03165-f005:**
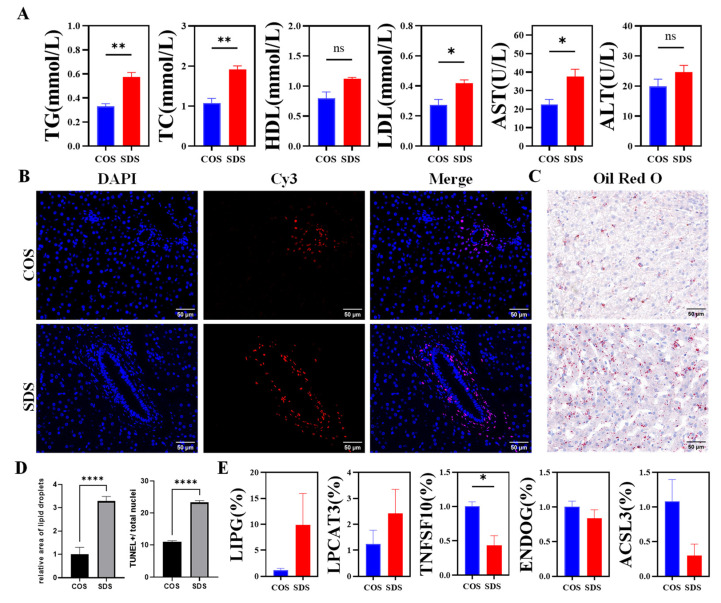
Sleep deprivation impacts lipid metabolism and induces hepatic injury. (**A**) Serum TG, TC, HDL, LDL, AST, and ALT levels. (**B**) TUNEL assay of liver tissue cells. Nuclei stained by Merge. Positive apoptotic cells are red. Blue is the nucleus. Scale bars, 50 μm. Magnification: ×40. (**C**) Representative images of liver sections stained with Oil red O. Scale bars, 50 μm. Magnification: ×40. (**D**) Quantitative data for the percentage of TUNEL-positive cells and the positive area of the Oil red O staining in the COS and SDS groups. (**E**). mRNA expression of LIPG, TNFSF10, LPCAT3, ENDOG, and ACSLE in the liver. All data are shown as the mean  ±  SEM. * *p*  <  0.05, ** *p*  <  0.01, **** *p*  <  0.0001.

## Data Availability

All data generated or analyzed during this study are included in this article, and materials are available from the authors upon reasonable request.

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
