# Peer review of "Transcriptome-Based Revelation of the Effects of Sleep Deprivation on Hepatic Metabolic Rhythms in Tibetan Sheep (*Ovis aries*)"

_animals, 2024, doi:10.3390/ani14223165_

Round 1

Reviewer 1 Report

Comments and Suggestions for Authors

Sleep is a vital process, the quality of which affects all areas of the body's activity. Sleep disorders are associated with the development of many diseases, including depression, hypertension, obesity and neurodegenerative diseases. The circadian process determines not only the onset and duration of sleep, but also many other parameters; in fact, almost every process in the body is under circadian control. In this regard, the presented studies on the effects of sleep deprivation on hepatic metabolic rhythms in Tibetan sheep (Ovis aries) are relevant and very interesting.

The introduction contains enough information to understand the problem, all information is accompanied by relevant links. The study design is well thought out, materials and methods are presented in detail and clearly.

The results correspond to the objectives. It is shown that sleep deprivation changes the circadian rhythm of gene expression in the liver. The following comments and suggestions arose regarding the work.

1) The materials and methods state

Line 178-179: Raw sequencing data are available in the sequence read archive under the accession numbers.

However, the numbers themselves are not provided.

2) Line 272-275: A differential analysis of expressed genes between the COS and SD treatment groups was performed to further reveal the role of expressed genes in regulating physiological 273

processes in the sheep liver, yielding 511 DEGs, with 210 upregulated and 301 downregulated genes.

Explain in which group gene expression is activated and in which it is suppressed. Are there any genes that clearly differentiate between groups (activated in one, suppressed in another, in what time interval does this occur)

3) Line 337-339: Notably, sleep depri-vation disrupted normal sleep-wake rhythms in sheep liver, leading to disruption of lipid metabolism and apoptosis in sheep liver.

It is not clear why the conclusion was made that this leads to apoptosis in sheep liver

4) Line 353-356: … suggesting that liver- mediated physiological processes have unique temporal structures for each individual, with differences in the temporal organization of the liver between nocturnal (mice) and diurnal (sheep) animals.

In discussions of the results, the main emphasis is placed on the results obtained earlier on mice, while in open sources there is now a lot of information related to circadian rhythms, sleep and liver function/condition processes in humans or large animals with a daily cycle. It is necessary to compare the obtained results not only with mice (nocturnal animals), but also with diurnal mammals.

Author Response

尊敬的审稿人/教授:

非常感谢您对稿件的积极和建设性的意见和建议。我们已根据您的意见和建议对稿件进行了大量修改,所有细节如下。

主要问题:

第 1 点第 178-179 行:原始测序数据可在序列读取档案中的入藏号下获得。但是,没有提供数字本身。

回应:感谢您对我的稿件的宝贵评论,您的评论对我们非常重要。我们深刻理解 SCI 期刊对数据共享的要求,并将原始数据上传到 NCBI 数据库。由于本次试验相关数据量较大,数据仍在上传中,目前尚未获得入藏号。预计完成上传大约需要 3 天时间。数据上传成功后,我们将立即联系编辑并补充此入藏号。

第 2 点第 272-275 行:对 COS 和 SD 处理组之间的表达基因进行差异分析,以进一步揭示表达基因在调节绵羊肝脏生理过程中的作用,产生 511 个 DEGs,其中 210 个上调和 301 个下调基因。解释哪个组基因表达被激活,哪个组基因表达被抑制。是否有任何基因可以明确区分各组(在一个组中激活,在另一个组中被抑制,这种情况发生在什么时间间隔内)

回应:谢谢你的提醒。在手稿中我们没有明确表达结果,稿件中出现的上调和下调基因与对照组 (COS) 相比都是治疗组 (SDS) 中上调和下调的基因,具体来说,即 210 个基因在 SDS 组被激活,在 COS 组中被抑制,另有 301 个基因在 COS 组被激活,在 SDS 组被抑制, 这些上调和下调基因的部分显示在“补充材料”中的图 S3A 和表 S2 中。同时,我们还更正了手稿的这一部分(第 289 - 291 行)。

第 3 点第 389-390 行:值得注意的是,睡眠不足扰乱了绵羊肝脏的正常睡眠-觉醒节律,导致绵羊肝脏脂质代谢中断和细胞凋亡。目前尚不清楚为什么得出这会导致绵羊肝脏凋亡的结论

回应:感谢您对我们稿件的关注和询问,这对我们具有重要意义。首先,我们承认我们手稿中的句子缺乏严谨性。在这项研究中,我们对睡眠剥夺后的肝脏进行了 TUNEL 染色,并量化了所得染色。TUNEL 染色是一种原位末端转移酶标记技术,可准确反映细胞凋亡最具特征性的生化和形态学特征。该实验的结果显示,睡眠剥夺后藏羊肝脏内 TUNEL 阳性细胞显著增加,表明睡眠剥夺导致肝细胞凋亡升高。此外,我们还对手稿中的这一部分进行了修订(第 348 - 349 行)。

第 4 点第 353-356 行:...表明肝脏介导的生理过程对每个个体都有独特的时间结构,夜间(小鼠)和昼夜(绵羊)动物之间的肝脏时间组织存在差异。在对结果的讨论中,主要强调早期在小鼠身上获得的结果,而在开源中,现在有很多与人类或具有日常周期的大型动物的昼夜节律、睡眠和肝功能/状况过程相关的信息。不仅需要将获得的结果与小鼠(夜行动物)进行比较,还需要与昼夜哺乳动物进行比较。

回应:感谢您对我们稿件的宝贵建议。这对改进我们的手稿非常有帮助。我们添加并比较了关于人类和其他日间活跃的大型动物的相关报告,这些报告可在手稿第 435 – 437 行和第 472 – 475 行中找到。

Reviewer 2 Report

Comments and Suggestions for Authors

Review on the manuscript titled “Transcriptome-based revelation of the effects of sleep deprivation on hepatic metabolic rhythms in Tibetan sheep (Ovis aries) by Chen et al., 2024

                The authors address the impact of Sleep deprivation (SD) on the hepatic metabolic rhythms.

In the introduction the authors underline the epidemiologic problem of rapid sleep duration contraction in Chinese human population from 8.5 to 7h during last decade, underscoring the problems of insufficient sleep duration. Also, the authors report “The liver, serving as the metabolic hub in the human body, plays a crucial role in various aspects of metabolism including carbohydrate, fat, and protein metabolism” in their problem statement/object justification.

The study employs Tibetan sheep species for sampling. It comprises 60 male Tibetan lambs 6 months old, 30 species control (COS) and 30 ones affected (SDS).

In the course of the study the authors arranged: a) 2.4 Histological analyses; b) 2.5 Measurement of Serum Biochemical panel; c) RNA-seq samples.

Figure 1 depicts the experimental design of the study during 7 days of experiment. The major object of study was the transcriptome profiles of the samples. The transcriptome available body of genes pool in both studies was approximately equal around 21000. The authors outlined the circadian genes (CG) as oscillating ones (3.1) in each group and the amount of CGs were approximately equal (Fig. 2).

The authors assessed the GO categories/pathways where enriched CG genes number was employed (Fig 2E) along with the polar plot indicating the aberrant CG profile in the affected group.

The next step was cluster analysis of CGs (Fig. 3) underscoring n alteration in the diurnal pattern of rumen metabolism (Fig. 3). The consequent analysis of DEG genes between groups (Chapter 3.3; Suppl. Fig S3; Fig. 4) revealed altered lipid metabolism in liver samples between the groups.

The consequent chapters: “3.4. The Impact of SD on Serum Biochemical Indicators, Lipid Metabolism, and Apoptosis in Tibetan Sheep Liver” revealed the impacts lipid metabolism and initiating hepatic injury (Fig. 5). The authors provided serum and tissue images evidence in the figure 5.

After rather meaningful discussion on the results obtained, where the authors underline certain genes and pathways, the authors concluded on the negative consequences on the SD.

While the manuscript is well designed and vocally sound, there are some points to mention:

a)      The authors should deposit their transcriptome data in public repository;

b)      The authors should provide the Supplementary with RNA-seq data statistics on each sample (coverage, quality score, etc.)

c)       It’s still not clear what strategy was employed for choosing RT-qPCR genes. It would be beneficial to characterize the sampling procedure and genes in a corresponding chapter.

d)      It is not clear why there are 2 Venn diagrams (Fig. 2B and Fig. S2), and why is there a difference between them? Please, elaborate in the captions. The same with Fig. 2D and Figure S2B.

e)      “ and genes with a fold difference >1.5 and P-value < 0.05 were selected as differentially expressed genes (DE)” – adjusted pvalue; or use FDR (in some figures also)

f)       “and proteins exhibiting significant circadian rhythmic changes had an ADJ.P < 0.05 [43]” - did you perform protein analysis on these, or were it actually gene expression measurements? Then ‘proteins’ should be replaced by ‘genes’;

g)      Some elaboration on JTK procedure is recommended to arrange in a supplementary.

h)      Actually, there are circadian CLOCK genes, which dictate the diurnal rhythms in the whole organism (some hormones, etc). The diurnal metabolism is maintained via this clock system organism wise. So is it reasonable to suppose that hepatic system maintains some independent diurnal rhythms? I am aware the authors trace the consequences of diurnal rhythm changes within hepatic system, but it’d be elaborated within diurnal rhythm system.

i)        ‘oscillating genes’ <> ‘circadian genes’.

j)        Figure S1: were these data employed in the analysis? Could the caption be more elaborated (e.g., for some conclusion herein)?

Author Response

尊敬的审稿人/编辑:

我们非常感谢您对我们稿件的有益评论和建议!我们已根据您的建议仔细修改了稿件,并尽可能完整地回答了您提出的每一点。请参阅以下逐项回复。

再次非常感谢!

一般评价:

第 1 点作者应将其转录组数据存放在公共存储库中

回应:感谢您对我的手稿的宝贵评论,您的评论对我们来说非常重要。我们深刻理解 SCI 期刊对数据共享的要求,并将原始数据上传到 NCBI 数据库。由于本次试验相关数据量较大,数据仍在上传中,目前尚未获得入藏号。预计完成上传大约需要 3 天时间。数据上传成功后,我们将立即联系编辑并补充此入藏号。

第 2 点作者应提供每个样品的补充 RNA-seq 数据统计数据(覆盖率、质量评分等)

回应:谢谢你提醒我。我们将在补充材料中提供详细的 RNA-seq 数据统计数据(表 S2),以证明我们结果的准确性和真实性,我们还在手稿的第 215-216 行描述了相关内容。

第 3 点目前尚不清楚采用什么策略来选择 RT-qPCR 基因。在相应的章节中描述采样程序和基因将是有益的。

回应:谢谢你提醒我。我们从差异表达基因中随机选择 5 个基因 (TNFSF10 、 ENDOG 、 ACSL3 、 LIPG 和 LPCAT3 ),进行 RT-qPCR 验证,探讨 SD 对藏羊肝脏基因表达的影响。同时,在手稿的相应部分(第 356 行)中补充了相关描述。

第 4 点目前尚不清楚为什么有 2 个维恩图(图 2B 和图 S2),为什么它们之间存在差异?请在标题中详细说明。图 2D 和图 S2B 相同。

回应:谢谢你的提醒。图 1 的维恩图S2A 基于 ADJ 基因。COS 组和 SDS 组中的 P<0.01,以及图 2 的两个聚类热图S2B 基于 ADJ 的表达水平。P<0.01 基因,并按相同顺序绘制。这部分图片的目的是进一步直观地表达睡眠剥夺后肝脏表达基因节律的紊乱。然而,图 2B 的维恩图是根据 ADJ 绘制的。COS 组和 SDS 组的 P<0.05 基因,以及图 2D 的热图与 COS 组和 SDS 组具有昼夜节律的基因作图,两个热图中包含的基因不同。为了更好地区分和表达图片的含义,我们将图 2 的标题改为“睡眠剥夺影响藏羊肝脏中表达的基因的昼夜节律”,并将 Fig.S2 到“睡眠剥夺破坏了藏羊肝脏基因的昼夜节律”。

第 5 点“和倍数差异为 >1.5 和 P 值< 0.05 的基因被选为差异表达基因 (DE)”——调整后的 pvalue;或使用 FDR(在某些数字中也是如此)

回应:谢谢你的提醒。我们非常同意您的建议,即我们将文章中的数据阈值更改为 FDR<0.05,并重新修改了结果的绘图和描述,如手稿的第 283-300 行和第 331-333 行以及补充材料的图 S3 所示。

第 6 点“表现出显着昼夜节律变化的蛋白质具有 ADJ。P < 0.05 [43]“ - 您是否对这些进行了蛋白质分析,或者实际上是基因表达测量?那么 'proteins' 应该被 'genes' 取代

回应:谢谢你的提醒。我们在新的手稿中更正了它。

第 7 点建议对 JTK 程序进行一些详细说明,以补充安排。

回应:谢谢你的好建议。我们在补充材料中已经详细解释了这个 JTK 程序。

第 8 点实际上,有昼夜节律 CLOCK 基因,它决定了整个生物体(一些激素等)的昼夜节律。昼夜代谢是通过这个时钟系统有机体来维持的。那么,假设肝脏系统保持一些独立的昼夜节律是否合理呢?我知道作者追踪了肝脏系统内昼夜节律变化的后果,但它会在昼夜节律系统中详细说明。

回应:非常感谢您的建议。这对我来说意义重大。外周时钟器官(如肝脏)的昼夜节律不仅受位于下丘脑的称为视交叉上核 (SCN) 的主时钟调节,还受各种其他因素的影响,包括光照、饮食模式、身体活动和睡眠-觉醒周期。这些因子通过不同的机制调节肝脏生理功能,从而影响新陈代谢、免疫反应和疾病易感性。研究表明,持续暴露在光线下会破坏肝脏正常的外周时钟节律,从而导致代谢紊乱[1]。当喂食高脂肪饮食时,Per1 敲除小鼠对肥胖的易感性降低,表明 Per1 在调节食物摄入对肝脏昼夜节律的影响中起关键作用[2]。肝脏的时钟基因表达受睡眠剥夺和不规则睡眠周期的影响。在一项研究中,发现急性睡眠剥夺会诱导外周血单核细胞 (PBMC) 中关键时钟基因表达的变化[3]。

 Point 9. ‘oscillating genes’ <> ‘circadian genes’.

Response: Thank you very much for your suggestion. In the whole paper, we have unified the usage and used oscillating genes, which have been modified and marked in the manuscript.

Point 10.Figure S1: were these data employed in the analysis? Could the caption be more elaborated (e.g., for some conclusion herein)?

Response:Thanks for your nice suggestion.Figure S1 shows that the animals in the test group and the control group were kept in the same environment (temperature and humidity), so as to exclude experimental errors due to environmental factors.

  1. Daisuke, Y., et al., 小鼠肝脏和白色脂肪组织中的外周昼夜节律被恒定光减弱,并通过限时进食恢复。PLOS ONE,2020 年。15(6):第 e0234439 页。
  2. Ge, W., et al., 昼夜节律 PER1 通过调节 PER1-PKA 对胆汁酸合成酶磷酸化的调节来控制每日脂肪吸收。脂质研究杂志, 2023.64(6):第 100390 页。
  3. Fagiani, F. 等人,昼夜节律的分子调控及其对生理学和疾病的影响。信号转导和靶向治疗,2022 年。7(1):第 41 页。

Round 2

Reviewer 1 Report

Comments and Suggestions for Authors

 Accept in present form

Author Response

请参阅附件

第 1 点。第 178-179 行:原始测序数据可在序列读取档案中的入藏号下获得。但是,没有提供数字本身。

回应:感谢您对我的稿件的宝贵评论,您的评论对我们非常重要。我们深刻理解 SCI 期刊对数据共享的要求,并已将原始数据上传到 NCBI 数据库(入藏号:PRJNA1177242),并在本次上传中修改了稿件的相关部分(第 180-181 行)。

第 2 点。作者应提供每个样品的补充 RNA-seq 数据统计数据(覆盖率、质量评分等)

 回应:感谢您对我的稿件的宝贵评论,您的评论对我们非常重要。我们提供每个样品的 RNA-seq 数据统计数据补充(表 S2)。
